# HSPC-Net: A hierarchical shape-preserving completion network for machine part point cloud completion

**Yuchao Jiang, Honghui Fan** *, **Hongjin Zhu**

Jiangsu University of Technology, Changzhou, Jiangsu, China

* fanhonghui@jsut.edu.cn

## Abstract

With the continuous advancement of 3D scanning technology, point cloud data of mechanical components has found widespread applications in industrial design, manufacturing, and repair. However, due to limitations in scanning precision and acquisition conditions, point cloud data often exhibit sparsity and missing information. This issue is particularly challenging when dealing with mechanically complex geometric shapes, where the missing portions frequently contain crucial details, posing significant difficulties for data completion. To effectively recover these missing parts while maintaining the accuracy of both global morphology and local details, this paper proposes a Hierarchical Shape-Preserving Completion Network (HSPC-Net). This approach integrates a multi-receptive field Transformer with a cross-modal geometric information fusion strategy, enabling the precise restoration of local details of mechanical components at multiple scales. Additionally, it leverages 2D image information to assist in the completion of 3D point clouds, significantly enhancing completion accuracy and robustness. Experimental results on ShapeNet and mechanical component point cloud datasets demonstrate that HSPC-Net outperforms existing state-of-the-art methods in terms of completion accuracy, structural consistency, and detail recovery.

## Introduction

With the rapid development of 3D scanning technology, point cloud data has been widely adopted in various domains, including industrial inspection, manufacturing, and robotic perception. In particular, the design, fabrication, and maintenance of mechanical components rely heavily on accurate 3D representations, where point clouds provide detailed geometric information of objects [1]. However, in practical acquisition scenarios, point clouds often suffer from sparsity and missing regions due to sensor limitations, occlusions, and environmental interference. This issue is especially critical in the context of mechanical parts, where the missing regions frequently contain essential geometric structures, such as edges, holes, and small-scale features. These geometric discontinuities not only compromise visual completeness but also significantly affect downstream tasks, thereby raising considerable challenges in accurate and structurally consistent completion [2,3].

**Data availability statement:** Our dataset has now been uploaded to Figshare: URL: https://figshare.com/articles/dataset/Mechanical_Part_Point_Cloud_Completion_Dataset/29602997?file=56388143

DOI: https://doi.org/10.6084/m9.figshare.
29602997

**Funding:** The author(s) received no specific funding for this work.

Completing mechanical component point clouds is particularly challenging due to several intrinsic factors. First, the geometric structures involved are often highly intricate, which makes the accurate reconstruction of missing regions difficult [4]. Second, due to the unstructured and unordered nature of point clouds, it is essential to maintain a delicate balance between the preservation of global morphology and the recovery of local details [5]. Fine features such as sharp corners, boundaries, and thin structures must be reconstructed precisely without compromising the coherence of the overall shape. Third, point clouds acquired through depth sensors or LiDAR are inherently sparse and noisy, and integrating complementary modalities such as 2D images introduces additional challenges. Although cross-modal data (e.g., RGB or depth images) can provide valuable geometric priors, effective fusion remains difficult due to discrepancies in data distribution, alignment errors, and the risk of introducing noise [6,7].

Existing methods attempt to address these issues from various perspectives. Some approaches focus on local feature refinement to enhance detail recovery but often lose sight of the overall structural integrity. Conversely, global shape modeling methods based on deep learning architectures can reconstruct coarse object shapes effectively but generally fail to preserve fine geometric attributes. Other works employ generative models to learn the distribution of complete shapes from large datasets and generate plausible completions. However, when applied to mechanical components, such methods may suffer from structural artifacts and insufficient fidelity due to the high complexity of the geometries involved. Moreover, cross-modal fusion strategies incorporating 2D images, depth maps, or RGB inputs have also been proposed to enrich 3D representations, yet challenges remain in achieving precise and stable integration of multimodal features. As a result, existing solutions are often limited in terms of completion accuracy, structural consistency, and robustness when facing real-world mechanical part scenarios.

To bridge these gaps, we propose a Hierarchical Shape-Preserving Completion Network (HSPC-Net), which incorporates hierarchical feature learning and cross-modal geometric fusion to improve the quality and consistency of point cloud completion. The core idea is to leverage a multi-receptive-field Transformer architecture that progressively encodes geometric features from local to global scales while introducing shape consistency constraints to stabilize the learning process. Furthermore, 2D image cues are utilized through a cross-modal Transformer to guide and refine 3D completion, enabling more accurate and robust recovery of complex structures.

Our contributions are as follows:

1. We introduce a Hierarchical Structure-Aware Completion method, capable of accurately restoring local details of mechanical components at multiple scales, while ensuring the consistency of the global structure.
2. We propose a Cross-Modal Geometric Fusion strategy, utilizing 2D image information to assist in the completion of 3D point clouds, further improving the precision and robustness of the completion results.
3. Experimental validation on the ShapeNet public dataset and a custom-built mechanical component point cloud dataset demonstrates that HSPC-Net significantly outperforms existing key methods in terms of point cloud completion accuracy and structural consistency, showcasing its superiority in practical applications. In particular, we select five state-of-the-art (SOTA) baselines for comparison, all of which represent the best-performing approaches reported in the literature across various paradigms.

## Related works

### Feature engineering

In the field of feature engineering for current deep learning-based point cloud completion algorithms, most researchers focus on two advanced algorithms: PointNet and DGCNN (Dynamic Graph Convolutional Neural Networks), both of which have demonstrated excellent performance in capturing the geometric and structural features of point cloud data. Table 1 summarizes and compares several key studies in point cloud completion, focusing on their feature engineering methods.

Xing et al. [8] employed the advanced DGCNN framework, specifically designing multi-level and sequential pooling DGCNN for segmenting the mining face point cloud, successfully capturing the key feature of the mining roof line. Regarding feature engineering for point cloud completion algorithms, some scholars have conducted survey-based studies. Diab et al. [9] investigated the state-of-the-art deep learning models, categorizing them based on data structures and evaluating their performance. Using benchmark datasets, they found that convolutional neural networks and their variants performed the best in remote sensing applications, being lightweight and effectively extracting key feature information. Furthermore, Xu et al. [10] proposed a feature fusion-based single-tree point cloud completion method, employing PointNet to extract global features and EdgeConv (Edge Convolution) to extract local features, which were then integrated and processed via FoldingNet to generate a complete point cloud. Compared to other methods on open-source datasets, this approach significantly improved Chamfer Distance (CD) and Earth Mover's Distance (EMD), effectively addressing the tree point cloud completion problem. Similarly, to improve point cloud task performance, Li et al. [11] introduced a deep autoencoder, where the encoder utilizes the PointNet framework to densely capture global features while incorporating a spectral graph learning module to infer spatial information, achieving advanced performance for point cloud learning tasks.

From the above studies, it is evident that both PointNet and DGCNN exhibit significant performance in feature engineering for point cloud tasks. However, when applied to the specific task of mechanical component point cloud completion, they face certain challenges. Mechanical parts typically have complex topologies and large-scale variations, making it difficult for local features to comprehensively represent global structure. Specifically, while DGCNN is powerful, its high computational complexity and limitations in handling long-range dependencies restrict its application in mechanical component point cloud completion tasks. On the other hand, although PointNet excels at global feature extraction, its ability to capture and process local features is insufficient when dealing with the complex topology and large-scale variations of mechanical components. This limitation can result in poor completion performance, failing to meet the accuracy requirements for mechanical component point cloud completion. In response, Wang et al. [12] proposed a dual-branch

**Table 1. Comparison of feature engineering methods in point cloud completion.**

| Method | PointNet | DGCNN | Dual-Branch Network | Feedback Network |
|---|---|---|---|---|
| Xing et al. [8] | | ✓ | | |
| Diab et al. [9] | ✓ | ✓ | | |
| Xu et al. [10] | ✓ | | ✓ | |
| Li et al. [11] | ✓ | | | |
| Wang et al. [12] | | | ✓ | |
| Yan et al. [13] | | ✓ | | ✓ |

network for shape completion, where the first branch synthesizes the complete object while preserving details, and the second branch reconstructs the original partial input. Both branches share a feature extractor to learn global features. However, although this method does not directly target feature extraction for mechanical component point cloud tasks, leading to some deviation in performance, the dual-branch feature extraction network provides a valuable idea for this study. Additionally, Yan et al. [13] proposed a novel feedback network for point cloud completion, which effectively refines current features by redirecting subsequent fine-grained features. This network first uses a hierarchical graph network to generate a rough shape, then cascades multiple feedback-aware completion blocks and recursively expands them over time. The feedback connections improve the current shape generation by utilizing fine-grained features, and the point-cross Transformer, designed to address dimensional mismatch issues, significantly enhances the effectiveness of feature engineering in point cloud tasks. However, the multiple feedback-aware completion blocks also introduce substantial computational complexity, which provides an important insight for this study in feature extraction.

Inspired by the aforementioned dual-branch feature extraction network and feedback network, this paper considers the use of a multi-receptive field Transformer encoding method. This approach effectively captures local geometric details and global structural information of point clouds at different scales, enabling precise recovery of the complete morphology of mechanical components. Moreover, by incorporating shape consistency constraints, it can further reduce structural deviations in point clouds under different viewpoints or poses, ensuring that the completed point clouds exhibit high geometric consistency.

## Point cloud completion

Another core aspect of deep learning-based mechanical component point cloud completion algorithms lies in the point cloud completion itself, following feature engineering, which is the ultimate goal of the research. Table 2 summarizes and compares several key studies in point cloud completion, focusing on their methods and innovations.

A common issue in point cloud tasks is the presence of holes. To address this, Yang et al. [14] proposed a portable, non-contact 3D measurement system for dairy cows using

**Table 2. Comparison of point cloud completion methods.**

| Method | Spline Curve Filling | Hierarchical Network | Topology Aware | Point Deconvolution | Encoder Decoder | Proxy Mechanism |
|---|---|---|---|---|---|---|
| Yang et al. [14] | ✓ | | | | | |
| Hu et al. [15] | | ✓ | | | | |
| Zhou et al. [16] | | ✓ | | | | |
| Xin et al. [17] | | | ✓ | | | |
| Tang et al. [18] | | | ✓ | | | |
| Peng Xiang et al. [19] | | | | ✓ | | |
| Wang et al. [20] | | | | | ✓ | |
| Cai et al. [21] | | | | | | ✓ |
| Li et al. [22] | | | | | | ✓ |
| Chen et al. [23] | | | | | | ✓ |
| Wu et al. [24] | | ✓ | | | ✓ | |
| Duan et al. [26] | | | ✓ | | ✓ | ✓ |
| An et al. [25] | | ✓ | | | | |
| An et al. [27] | | ✓ | | | | |

smartphones. By employing photogrammetry techniques, they reconstructed 3D point cloud scenes of cows and applied spline curve methods to fill the holes in the point cloud data.

Additionally, numerous researchers have focused on generating high-quality 3D objects in point cloud tasks. For instance, Hu et al. [15] introduced a hierarchical shape-preserving network (HSPN), which cleverly integrates branch predictors and hierarchical attention pipelines. This network efficiently generates initial point cloud data while intelligently identifying and completing missing parts in images, resulting in a high-quality and complete 3D brain point cloud model. Similarly, to restore the complete shape of 3D objects with high quality, Zhou et al. [16] proposed a novel point cloud completion method called SeedFormer. This method enhances detail preservation and recovery in point cloud completion by introducing Patch Seeds, which capture global structure and retain local pattern information. By combining seed features, SeedFormer restores fine-grained details of the complete point cloud through a multi-scale approach.

However, the unordered nature of point clouds can also affect the generation of high-quality 3D shapes. To address this, Xin et al. [17] proposed treating completion as a point cloud deformation process and designed PMP-Net++ (Point Moving Path-Network++), a neural network that mimics the behavior of a "bulldozer." This network moves each point in the incomplete input to the corresponding complete point cloud while minimizing the total movement path distance. Similarly, Tang et al. introduced LAKe-Net (Localizing Aligned Keypoints-Network) [18], a novel topology-aware point cloud completion model. It works by aligning keypoints and adopting a novel keypoint-skeleton-shape prediction approach, which addresses the topological missing data problem in point cloud completion by locating aligned keypoints, generating surface skeletons, and refining the shape.

At the same time, point cloud completion often faces issues such as discreteness and local unordered predictions. To tackle these problems, Xiang et al. [19] proposed Snowflake Point Deconvolution (SPD), which simulates point cloud growth in a flake-like manner by progressively splitting parent points into child points. SPD introduces a skip-connection Transformer to learn the optimal local splitting patterns. This Transformer utilizes attention mechanisms to summarize previous layer splitting patterns and guide current layer splitting, generating compact and orderly local point clouds that accurately reveal the local structural features of 3D shapes and predict high-precision geometric details. Similarly, Wang et al. [20] introduced a novel encoder-decoder structure with a Transformer-based model. This method improves point cloud processing capabilities through feature extraction, alternative neighborhood pooling operations, and upsampling steps. The Transformer architecture further enhances the generalization ability of the approach. Another solution was proposed by Cai et al. [21], who learned a unified and structured latent space by encoding partial and complete point clouds. They mapped a series of related partial point clouds to multiple complete shapes and occlusion codes, fused these codes to obtain representations in the unified latent space, and implemented structural ranking regularization, latent code exchange constraints, and distribution supervision for related partial point clouds, thereby achieving better partial-complete geometric consistency and shape completion accuracy.

In addition to the above methods, Li et al. [22] proposed a new point cloud completion method called Proxy-Former. This method divides the point cloud into existing (input) and missing (to-be-predicted) parts, and employs a proxy mechanism for information communication to achieve precise prediction of the missing parts. Specifically, Proxy-Former designed a missing part-sensitive Transformer and proxy alignment mechanism, enhancing the sensitivity of predicted points to the features and positions of the missing parts.

From the above studies, it is evident that existing point cloud processing solutions perform excellently across various tasks. However, unfortunately, they are not fully applicable to

mechanical component point cloud processing. This limitation arises from the unique characteristics of mechanical components. The surfaces of mechanical parts consist of edges and planes and contain multiple complex geometries. Point cloud data from these parts is rich in regular structures, and the overall shape is structured and precise. These characteristics present challenges for existing methods when dealing with mechanical components, including complex geometric structures, severe data loss, and inadequate generalization capabilities of current methods.

Chen et al. [23] proposed a new point cloud completion architecture called Anchor-Former, which dynamically captures object region information using pattern-aware discriminative nodes (anchor points). This method addresses the problem of global feature vectors struggling to fully represent the diverse patterns of objects. By learning anchor points and applying modulation schemes, AnchorFormer transforms sparse points and standardized 2D grids into fine 3D structures, achieving high-quality object pattern reconstruction. While this approach has high memory consumption and computational complexity, the conversion between 2D networks and 3D structures offers valuable insights for this study.

Recent advances have addressed point cloud completion in sparse and few-shot scenarios. Wu et al. [24] proposed FSC, a dual-branch framework capable of completing shapes from as few as 64 points, showing strong generalization across categories. Duan et al. [26] introduced T-CorresNet, which employs a spherical template and correspondence pooling to guide coarse-to-fine shape generation. An et al. [25,27] explored few-shot 3D segmentation through prototype-based correlation optimization and multimodal fusion using text and images. While these methods perform well on common object datasets, their applicability to mechanical components remains limited due to the need for accurate reconstruction of intricate geometric structures, the absence of suitable templates, and the lack of auxiliary modalities in industrial settings.

## Proposed method

### Overall framework

In view of the complexity of the point cloud completion task of mechanical parts, A Hierarchical Shape-preserving Completion Network (HSPC-Net) is proposed, which makes full use of 3D geometric features and improves the accuracy and structural consistency of high point cloud completion. HSPC-Net is composed of two core modules: hierarchical structure sensing point cloud completion and cross-modal geometry information fusion, which are used to restore the complex geometry of mechanical parts, and enhance the effect of point cloud completion by using multi-source data. The overall framework of HSPC-Net is shown in Fig 1.

Most of the existing point cloud completion methods focus on the generation of global shapes or the completion of local details. In the complex mechanical parts scene, it is necessary to accurately restore small-scale features (such as edges and holes) and ensure the consistency of the overall structure to maintain the geometric integrity of the parts. Therefore, HSPC-Net adopts a hierarchical feature extraction strategy based on multi-field Transformer, obtains geometric information of different scales through progressive coding, and combines shape consistency constraint mechanism to suppress geometric deviation caused by changes in viewing angle or data noise. HSPC-Net further introduces cross-modal information interaction, and uses 2D visual information to guide the whole 3D complement process, so as to enhance the robustness of depth estimation and the rationality of spatial structure.

The point cloud completion process of HSPC-Net mainly includes four stages: input coding, hierarchical completion, cross-modal fusion and final reconstruction.

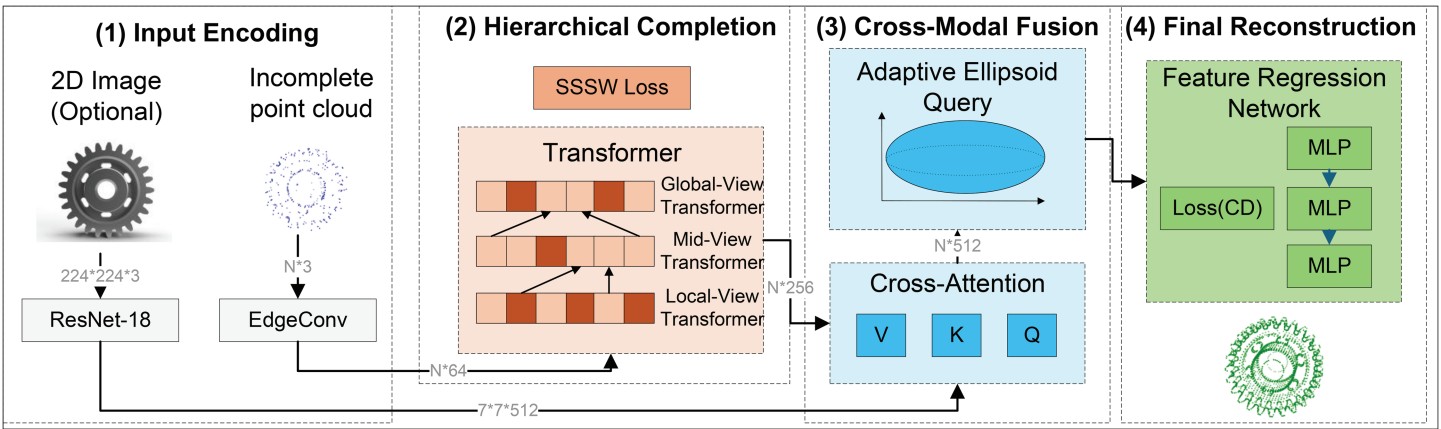

**Fig 1. The overall framework of HSPC-Net.**

1. Input Encoding: Incomplete point cloud data is first passed through local feature extraction module to obtain preliminary geometric representation, while 2D image data is feature encoded through convolutional neural network to extract depth-related information.

2. Hierarchical Completion, adopt multi-layer Transformer structure for feature extraction of point cloud, gradually expand the sensing field, and optimize the feature distribution of point cloud through shape consistency constraints to maintain geometric stability under different viewing angles.

3. Cross-Modal Fusion uses 2D-3D interactive module to introduce 3D structure information into the depth estimation process, and optimizes the selection of 3D domain information through adaptive ellipsoid query to improve the completion accuracy.

4. Final Reconstruction: Feature regression is carried out on the fused 3D features to generate high-precision point cloud completion results, and the quality of point cloud reconstruction is optimized by using Chamfer Distance constraint.

### Hierarchical structure perception point cloud completion

One of the fundamental challenges in point cloud completion lies in achieving a balance between preserving global geometric consistency and recovering fine-grained local details. Methods focusing solely on global constraints tend to oversmooth sharp features such as edges and corners, while those emphasizing only local structures may lead to noise amplification or structural disintegration due to the lack of contextual guidance. To address this issue, we propose a hierarchical structure perception point cloud completion framework. This framework employs a multi-receptive field self-attention mechanism to extract geometric features across different scales and integrates a shape consistency constraint to ensure the completed point cloud maintains stable and coherent geometry under varying perspectives or transformations.

**Multi-receptive field transformer structure.** In point cloud processing, the unordered nature of 3D points makes it challenging to apply standard convolution operations. Therefore, we adopt a Transformer architecture based on a self-attention mechanism to capture spatial dependencies, enabling joint encoding of both local geometric features and global structural information.

Let the input incomplete point cloud be denoted as

$$P = \{p_i \mid p_i \in \mathbb{R}^3, i = 1, \dots, N\}, \tag{1}$$

where $p_i$ is the $i$-th point and $N$ is the total number of points. The initial feature representation is defined by the point coordinates:

$$F^0 = P. \tag{2}$$

The feature extraction process is performed through a stack of $D$ Transformer layers with different receptive fields. The feature output of each layer is defined as:

$$F^d = \Phi_d(F^{d-1}), \quad d = 1, \dots, D \tag{3}$$

where $\Phi_d(\cdot)$ denotes the transformation function of the $d$-th Transformer layer, responsible for progressively extracting structural features at multiple scales.

To simultaneously capture local and global contexts, we introduce a multi-receptive field control mechanism within the self-attention computation. The self-attention operation is defined as:

$$\text{Att}(Q, K, V) = \text{softmax}\left(\frac{QK^\top}{\sqrt{d_k}} + M_d\right) V \tag{4}$$

where $Q$, $K$, and $V$ are the Query, Key, and Value matrices computed from the input features via learnable linear projections:

$$Q = F^{d-1} W_Q, \quad K = F^{d-1} W_K, \quad V = F^{d-1} W_V \tag{5}$$

with $W_Q, W_K, W_V \in \mathbb{R}^{C \times d_k}$ representing the projection matrices, $C$ the input feature dimension, and $d_k$ the dimension used for attention scaling.

To regulate the attention range at different layers, we define a receptive field mask $M_d$ as:

$$M_d(i, j) = \begin{cases} 0, & \text{if } |p_i - p_j| < r_d \\ -\infty, & \text{otherwise} \end{cases} \tag{6}$$

where $r_d$ denotes the receptive field radius at layer $d$. As $d$ increases, $r_d$ also increases, enabling shallow layers to focus on local features and deeper layers to aggregate global structural information.

The output of each Transformer layer is updated using a residual connection followed by layer normalization:

$$F^d = \text{LN}\left(\text{Att}(Q, K, V) + F^{d-1}\right) \tag{7}$$

where $\text{LN}(\cdot)$ represents the layer normalization operation to stabilize training and improve convergence.

**Shape consistency constraints.** While the multi-receptive field Transformer can extract features across hierarchical levels, it may still suffer from shape inconsistencies due to noise, sampling variations, or changes in viewpoint. To mitigate this issue, we introduce a Shape

Consistency Constraint (SCC), which enforces consistency of spatial feature distributions across different stages by aligning their second-order statistics.

Specifically, we adopt a Second-order Shape Selective Whitening (SSSW) loss to constrain the structural consistency of point cloud features. The loss is defined as:

$$L_{\text{SSSW}} = \sum_{d=1}^{D} \left( \left| V_s^d \odot O_d \right| + \left| V_t^d \odot O_d \right| \right) \tag{8}$$

where $V_s^d$ and $V_t^d$ denote the covariance matrices of the source and target features at layer $d$, respectively:

$$V_s^d = \frac{1}{N}(F_s^d)^\top F_s^d, \quad V_t^d = \frac{1}{N}(F_t^d)^\top F_t^d \tag{9}$$

Here, $F_s^d$ and $F_t^d$ are the feature matrices of the source and target point clouds at layer $d$.

$O_d$ is a shape-sensitive feature selection matrix, which identifies $k$ channels with the most significant shape differences. This matrix is defined by a threshold $\delta$ as:

$$O_d(i,j) = \begin{cases} 1, & \text{if } \left| V_s^d(i,j) - V_t^d(i,j) \right| > \delta \\ 0, & \text{otherwise} \end{cases} \tag{10}$$

By selectively aligning only the most morphologically sensitive components, the SSSW loss helps suppress unstable variations and enhances the robustness and structural fidelity of the completed point cloud. It works in conjunction with the primary point cloud reconstruction loss to guide the network toward learning feature representations that are both geometrically consistent and detail-preserving.

## Cross-modal geometric information fusion

In the aforementioned hierarchical structure sensing point cloud completion module, the multi-field Transformer structure is used to extract the local and global features of the point cloud, and shape consistency constraints are introduced to improve the geometric consistency of the completed point cloud. However, only relying on 3D shape information still has certain limitations. For example, when the input point cloud has a large area missing, only inferring the distribution of completion points from the already partial cloud will easily lead to shape distortion or local structure blurring. In addition, point cloud data usually comes from depth sensors, whose measurement errors may lead to inaccurate surface normal calculations, thus affecting the quality of completion. Therefore, cross-modal geometric information fusion is proposed in this section. The depth, edge and geometric prior information provided by 2D images are used for collaborative optimization with 3D point cloud features to improve the completion accuracy and stability.

**Cross-mode transformer interaction.** In the complement process, 2D images contain rich geometric clues, such as depth gradient, edge information and shape prior, while 3D point cloud data directly carries 3D spatial structure. Therefore, a Cross-Modal Transformer (CMT) is designed for information interaction between 2D and 3D. The core goal of this module is to establish semantically consistent Feature mapping between 2D and 3D through Adaptive Feature Alignment and Attention Weight Assignment. Thereby enhancing 3D morphological completion with 2D priors. Let the input 2D image features and 3D point cloud features be:

$$X_{2D} = \{x_i^{2D} \mid x_i^{2D} \in R^{C_{2D}}, i = 1, \dots, N_{2D}\} \tag{11}$$

$$X_{3D} = \{x_j^{3D} \mid x_j^{3D} \in R^{C_{3D}}, j = 1, \dots, N_{3D}\} \tag{12}$$

Where, $x_i^{2D}$ represents the $i$-th feature vector of 2D image, $x_j^{3D}$ represents the $j$-th feature vector of 3D point cloud, $C_{2D}$ and $C_{3D}$ are the number of channels of 2D and 3D features respectively, $N_{2D}$ and $N_{3D}$ are the corresponding feature points. When performing feature fusion between 2D and 3D, a Cross-Modal Attention Mechanism is used to define the 2D image features as Key and Value, and the 3D point cloud features as Query:

$$Q_{3D} = X_{3D} W_Q, \quad K_{2D} = X_{2D} W_K, \quad V_{2D} = X_{2D} W_V \tag{13}$$

Where, $W_Q, W_K, W_V \in R^{C_{2D} \times d_k}$ is a learnable parameter. Cross-modal attention is calculated as follows:

$$F_{\text{fusion}} = \text{softmax}\left(\frac{Q_{3D} K_{2D}^\top}{\sqrt{d_k}}\right) V_{2D} \tag{14}$$

Where $d_k$ is the scaling factor, the function is to stabilize the gradient update of the attention calculation. Finally, the 3D feature representation after fusion is updated as:

$$F_{3D'} = \text{LayerNorm}\left(F_{3D} + F_{\text{fusion}}\right) \tag{15}$$

The above calculation process realizes the propagation of 2D image geometry information to 3D point cloud, and the stability of information fusion is optimized with Transformer structure.

**Adaptive context optimization based on ellipsoid query.** Since the point cloud completion task involves the inference of geometric relationship of missing points, how to efficiently select suitable neighborhood points for feature extraction is crucial. The traditional Ball Query searches for neighborhood points within a fixed radius, but there are still some problems: if the density of point cloud is low, the fixed radius may lead to too few neighborhood points, which affects the contextual information integrity of the completion point; If the density of point cloud is high, ball query may introduce a large number of irrelevant points, increasing the computational redundancy; Mechanical parts usually have anisotropic geometric characteristics, and spherical query can not adapt to complex surface structures. To solve the above problems, a context optimization strategy based on Ellipsoid Query is proposed, and the query range is adjusted adaptively to improve the accuracy of high-point cloud feature extraction.

For any query point $p = (x, y, z)$, the query conditions based on ellipsoid constraints are defined as follows:

$$\frac{x'^2}{a^2} + \frac{y'^2}{b^2} + \frac{z'^2}{c^2} \leq 1 \tag{16}$$

Where, $x', y', z'$ are the transformed point cloud coordinates, and $a, b, c$ are the long axis and short axis of the ellipsoid respectively, whose sizes are adjusted adaptively according to the point cloud density. Let $N_{\text{local}}$ be the number of points in the neighborhood and $V$ be the volume of the local area, then the local point cloud density is:

$$\rho = \frac{N_{\text{local}}}{V} \tag{17}$$

Let $\alpha, \beta$ be hyperparameters, then define the adaptive adjustment rules for the long and short axes of the ellipsoid as:

$$a = \alpha \cdot \frac{1}{\rho}, \quad b = \beta \cdot a, \quad c = \beta \cdot a \tag{18}$$

When the local point cloud density is low, $a$ is increased to expand the search scope and improve the contextual information integrity of the completion point. When the point cloud density is high, $a$ is reduced to reduce redundant points and improve computing efficiency.

The loss of this part is represented by the cross-modal geometric fusion loss $L_{\text{cross - modal}}$, which is mainly used to optimize the information interaction between 2D-3D and ensure the reasonable distribution of the completed point cloud under the constraint of 2D image. The loss is expressed by the cross-modal characteristic consistency loss, which is mainly investigated as:

$$L_{\text{cross}_{\text{m}}\text{odal}} = \sum_{i=1}^{N_{\text{3D}}} \left\| F_{\text{3D},i} - F_{\text{fusion},i} \right\|^2 \tag{19}$$

Where, $F_{\text{3D},i}$ and $F_{\text{fusion},i}$ respectively represent the 3D original features and the fusion features after cross-mode Transformer calculation, which ensures the effective propagation of 2D image information to the 3D point cloud filling process.

## Objective function

The goal of the point cloud completion task is to recover the structure information of the missing point cloud and ensure the rationality of the overall shape and local details while minimizing geometric errors. In the HSPC-Net structure, the optimization of the completion results involves several interrelated objectives, including shape consistency, cross-modal information fusion, and the final point cloud reconstruction. Therefore, the objective function design adopts the Multi-Objective Optimization strategy. Second-order Shape Selective Whitening, SSSW), Cross-Modal Geometric Fusion Loss (SSSW) and Reconstruction Loss (SSSW) to ensure the robustness and accuracy of point cloud completion. In addition, since the contribution of each loss item may change with the task requirements, Fuzzy Mathematical Programming is introduced to realize adaptive loss weight adjustment, and improve the optimization efficiency and generalization ability of the network.

Let the final total loss function $L_{\text{total}}$ consist of three parts:

$$L_{\text{total}} = \lambda_1 L_{\text{SSSW}} + \lambda_2 L_{\text{cross - modal}} + \lambda_3 L_{\text{reconstruction}} \tag{20}$$

Among them, the point cloud reconstruction loss $L_{\text{reconstruction}}$ is used to measure the geometric error between the completed point cloud and the real point cloud, which is mainly calculated using Chamfer Distance (CD). Set $P$ and $Q$ as the completed point cloud and the target point cloud respectively, then $L_{\text{reconstruction}}$ is expressed as:

$$L_{\text{reconstruction}} = \sum_{p \in P} \min_{q \in Q} p - q^2 + \sum_{q \in Q} \min_{p \in P} q - p^2 \tag{21}$$

Because of the different contribution of different loss terms to the optimization objective, its weight coefficients $\lambda_1, \lambda_2, \lambda_3$ need to be dynamically adjusted. The traditional method usually adopts fixed weight, but in the actual optimization process, the importance of each loss item may change under different task scenarios. Therefore, Fuzzy Mathematical Programming (FMP) is introduced to adjust the loss weight adaptively and improve the flexibility of optimization. The fuzzy membership function of the target loss is set as:

$$\mu_{\lambda_k}(L_k) = \exp\left(-\frac{(L_k - L_k^0)^2}{2\sigma_k^2}\right), \quad k \in \{1, 2, 3\} \tag{22}$$

Where, $L_k^0$ is the initial loss value, $\sigma_k$ is the scale parameter of the loss change, $\mu_{\lambda_k}(L_k)$ reflects the importance of the loss term $L_k$, and the range is [0,1]. The weight update formula is as follows:

$$\lambda_k = \frac{\mu_{\lambda_k}(L_k)}{\sum\limits_{j=1}^{3} \mu_{\lambda_j}(L_j)} \tag{23}$$

This method can dynamically adjust the influence of loss terms and make the optimization process more robust.

## Experimental analysis

### Experimental setup

In order to comprehensively evaluate the performance of the proposed HSPC-Net in the point cloud completion task, this section provides a detailed description of the experimental setup, including the software and hardware environment, datasets, comparison methods, experimental metrics, and the experimental design.

The hardware configuration of the experimental environment includes a computer equipped with an AMD Ryzen 7 8-core 16-thread processor, 32GB of RAM, a GeForce RTX 4060 graphics card, and the Windows 11 operating system. The deep learning framework used is PyTorch 1.12.1, compatible with CUDA 11.6.

To comprehensively evaluate the performance of HSPC-Net, we adopt a two-stage training strategy. First, we perform general shape pre-training on the full ShapeNet dataset, which consists of over 220,000 3D CAD models across 3,135 categories, hierarchically organized according to WordNet. This pre-training phase enables the network to acquire general geometric priors, such as symmetry and topology, from diverse object shapes. In the second phase, we conduct domain-adaptive fine-tuning on a custom mechanical component point cloud dataset. This dataset includes 2,000 samples of industrial parts generated through laser scanning technology, such as spur gears, bevel gears, helical gears, worm gears, and worm wheels, which often exhibit sparse, incomplete, and noisy point clouds reflecting real-world manufacturing scenarios.

For evaluation, we select four representative ShapeNet categories, Airplane, Car, Laptop, and Chair, to report per-category and mean performance. We use the standard training/validation/test split for all datasets: 85% for training, 5% for validation, and 10% for testing. Specifically, for each selected category, we report results on the held-out 10% test set.

Five representative baseline methods in the current point cloud completion field were selected for comparison:

1. **Point Fractal Network (PF-Net)** [28] A point cloud completion method based on a hierarchical feature generation network. It estimates the missing point clouds through a multi-scale generation network and introduces multi-stage completion loss and adversarial loss to generate more realistic completed regions.
2. **Cascaded Refinement Network (CRN)** [12] A cascaded shape completion subnetwork that progressively refines rough point cloud completion results to preserve object details.
3. **Feedback Network (FBNet)** [13] Introduces a feedback mechanism that uses cross-time-step feedback connections to iteratively optimize the current features, enhancing the fine details of the completion.
4. **Hierarchical Shape-Perception Network (HSPN)** [15] A shape reconstruction network based on hierarchical attention pipelines and attention gate blocks. It aggregates local geometric features and internal representations to complete 3D point clouds from incomplete observations.
5. **Template-guided Correspondence Network (T-CorresNet)** [26] A template-guided completion framework that embeds a spherical prior and uses a correspondence pooling strategy to generate dynamic queries, followed by a FoldingNet-based decoder for coarse-to-fine point cloud refinement.

To thoroughly evaluate the point cloud completion results, three metrics are employed in this experiment:

1. **Chamfer Distance (CD, L2)** A lower CD value indicates better performance, as it reflects a higher geometric similarity between the completed point cloud and the target point cloud.
2. **Cascaded Refinement Network (CRN)** This metric measures the consistency between the generated point cloud and the target point cloud, emphasizing the accuracy of shape recovery. It is defined as the degree of overlap between the completed point cloud and the ground truth point cloud. A higher PCA value indicates that the completion result is closer to the target point cloud in terms of shape and structure.
3. **Structural Consistency Score (SCS)** This metric is used to evaluate the geometric consistency of mechanical component point cloud completion. It quantifies the ability of the completion result to maintain the structure by calculating the degree of match between the local and global morphology of the completed point cloud. It is defined as:

$$\text{SCS} = \frac{1}{N} \sum_{i=1}^{N} \cos(\theta_i) \tag{24}$$

Where $\theta_i$ represents the angle between the normal vector of the completed point cloud and the target point cloud at the $i$-th point. The higher the SCS, the stronger the geometric consistency of the completion point cloud.

## Ablation study of HSPC-Net

To further investigate the contribution of each module in HSPC-Net to point cloud completion performance, an ablation study was conducted. This study systematically removed the hierarchical structure perception module and the cross-modal geometric fusion module from HSPC-Net to observe the specific impact of these modules on point cloud completion performance. The primary goal of the ablation study was to verify the contribution of each

module to completion accuracy, geometric consistency, and shape recovery, thereby providing theoretical support for model optimization and design.

For the experiments, four categories—Airplane, Car, Laptop, and Chair—from ShapeNet were selected as the datasets. During the experiment, HSPC-Net was divided into three versions: the full version, the version without the hierarchical structure perception module, and the version without the cross-modal module. Each version was trained and evaluated using the same training data and evaluation metrics to ensure fairness in the experiment. The results of the experiments with the three versions of HSPC-Net are shown in Table 3.

As shown in the table, the full version of HSPC-Net outperforms the other two versions in overall performance, achieving the lowest CD, the highest PCA, and the highest SCS. This indicates that both modules play a significant role in enhancing point cloud completion accuracy, precision, and structural consistency.

In this study, the contributions of the hierarchical structure perception module and the cross-modal geometric fusion module were analyzed in depth. For the version without the hierarchical structure perception module, the CD increased from 0.023 to 0.031, the PCA decreased from 0.892 to 0.848, and the SCS also dropped from 0.915 to 0.883. These results suggest that the hierarchical structure perception module plays a crucial role in maintaining the geometric consistency and recovering details in the completed point cloud. This module effectively captures the relationship between local details and global morphology by extracting point cloud features at different scales through the multi-receptive field Transformer, thus playing a key role in local detail recovery and global shape consistency.

For the version without the cross-modal geometric fusion module, although the CD and SCS improved slightly, the PCA remained lower than the full model, decreasing from 0.892 to 0.874. This suggests that the cross-modal geometric fusion module plays an active role in enhancing 3D completion, particularly in the accuracy of shape recovery. Through the information propagation of the 2D-3D Transformer, this module effectively integrates 2D image information into the completion process, making the point cloud completion more aligned with the geometric structure of real objects. When both the hierarchical structure perception module and the cross-modal geometric fusion module are present, the completion performance is further improved.

## Comparative experiment on shapeNet dataset

To evaluate the performance of HSPC-Net in point cloud completion tasks, we conducted comparative experiments with five state-of-the-art methods. The experiment selected four categories—Airplane, Car, Laptop, and Chair—from ShapeNet to comprehensively assess the advantages of HSPC-Net in shape recovery, geometric consistency, and completion accuracy. The experimental results on the ShapeNet dataset are shown in Table 4.

From the experimental results, HSPC-Net demonstrates excellent performance in the point cloud completion tasks across all four categories in most cases. In terms of the CD metric, HSPC-Net achieves the best performance, particularly in the Airplane (0.023) and Chair (0.025) categories, clearly outperforming the other methods. In comparison, PF-Net

**Table 3. Ablation experiment results of HSPC-Net.**

| Method | CD | PCA | SCS |
|---|---|---|---|
| HSPC-Net (Full) | **0.023** | **0.892** | **0.915** |
| HSPC-Net (Without Hierarchical Module) | 0.031 | 0.848 | 0.883 |
| HSPC-Net (Without Cross-modal Module) | 0.028 | 0.874 | 0.895 |

**Table 4. Category-wise comparison of point-cloud completion methods on ShapeNet.**

| Method | Airplane | | | Car | | | Laptop | | | Chair | | |
|---|---|---|---|---|---|---|---|---|---|---|---|---|
| | CD | PCA | SCS | CD | PCA | SCS | CD | PCA | SCS | CD | PCA | SCS |
| PF-Net | 0.029 | 0.876 | 0.907 | 0.036 | 0.842 | 0.892 | 0.033 | 0.861 | 0.887 | 0.031 | 0.854 | 0.890 |
| CRN | 0.026 | 0.888 | 0.913 | 0.033 | 0.858 | 0.901 | 0.029 | 0.869 | 0.895 | 0.027 | 0.860 | 0.898 |
| FBNet | 0.024 | **0.892** | **0.918** | 0.032 | 0.864 | 0.903 | 0.030 | 0.870 | 0.898 | 0.028 | 0.863 | 0.902 |
| HSPN | 0.027 | 0.885 | 0.910 | 0.034 | 0.855 | 0.898 | 0.031 | 0.866 | 0.892 | 0.029 | 0.857 | 0.896 |
| T-CorresNet | 0.025 | 0.890 | 0.914 | 0.032 | 0.867 | 0.905 | 0.0289 | 0.875 | 0.902 | 0.026 | 0.868 | 0.902 |
| HSPC-Net | **0.023** | **0.892** | 0.915 | **0.031** | **0.874** | **0.909** | **0.028** | **0.882** | **0.912** | **0.025** | **0.874** | **0.905** |

and CRN perform poorly, with PF-Net exhibiting a CD of 0.029 in the Airplane category, indicating a large geometric error between the generated point cloud and the target point cloud. This can be attributed to PF-Net's tendency to introduce unnecessary geometric structures when handling missing regions, leading to deviations in the completion results. In contrast, HSPC-Net effectively avoids this issue by using hierarchical structure perception and cross-modal geometric information fusion, thus maintaining geometric consistency and reducing completion errors.

Notably, although HSPN improves over PF-Net and CRN to some extent (e.g., CD = 0.027 on Airplane), its performance remains inferior to FBNet and T-CorresNet. The latter achieves relatively low CD values (e.g., 0.025 on Airplane and 0.026 on Chair), but still slightly lags behind HSPC-Net.

Regarding the PCA metric, HSPC-Net's performance is comparable to that of FBNet across multiple categories, but it slightly outperforms FBNet overall, especially in the Laptop (0.882) and Car (0.874) categories, where HSPC-Net achieves higher accuracy. This suggests that HSPC-Net is more accurate in shape recovery. In contrast, PF-Net shows significant deviations in shape recovery, particularly in the Airplane category, where its PCA score is 0.876, failing to effectively restore the details of the target point cloud. T-CorresNet also performs favorably, with PCA scores of 0.890 (Airplane) and 0.868 (Chair), further validating the benefit of using structure-aware representations.

For the SCS metric, HSPC-Net exhibits stable performance, with SCS values of 0.915 and 0.912 in the Airplane and Laptop categories, respectively. Although slightly lower than FBNet, HSPC-Net still holds a clear advantage, demonstrating strong geometric consistency. In the Car category, FBNet slightly outperforms HSPC-Net (0.903 vs. 0.909), which may be due to FBNet's stronger completion capability in high-density regions. T-CorresNet also shows competitive SCS scores (e.g., 0.914 on Airplane, 0.902 on Chair), further confirming its reliability. On the other hand, HSPN performs weaker than CRN across all categories, reaffirming the necessity of better structure modeling.

Fig 2 shows the comparison of actual point cloud completion effects of different methods on the ShapeNet dataset. To be more representative, we selected the Airplane category. To keep the figure concise, we only plotted a portion of the points. It can be seen that as the proportion of input point clouds to the complete point clouds gradually increases, the completion effects of all methods improve. However, HSCP-Net outperforms the comparison methods when the input point cloud proportion is 10%, especially under the same conditions, PF-Net shows poor completion results.

Overall, HSPC-Net is an effective point cloud completion method with high application potential, particularly suited for scenarios requiring precise geometric recovery.

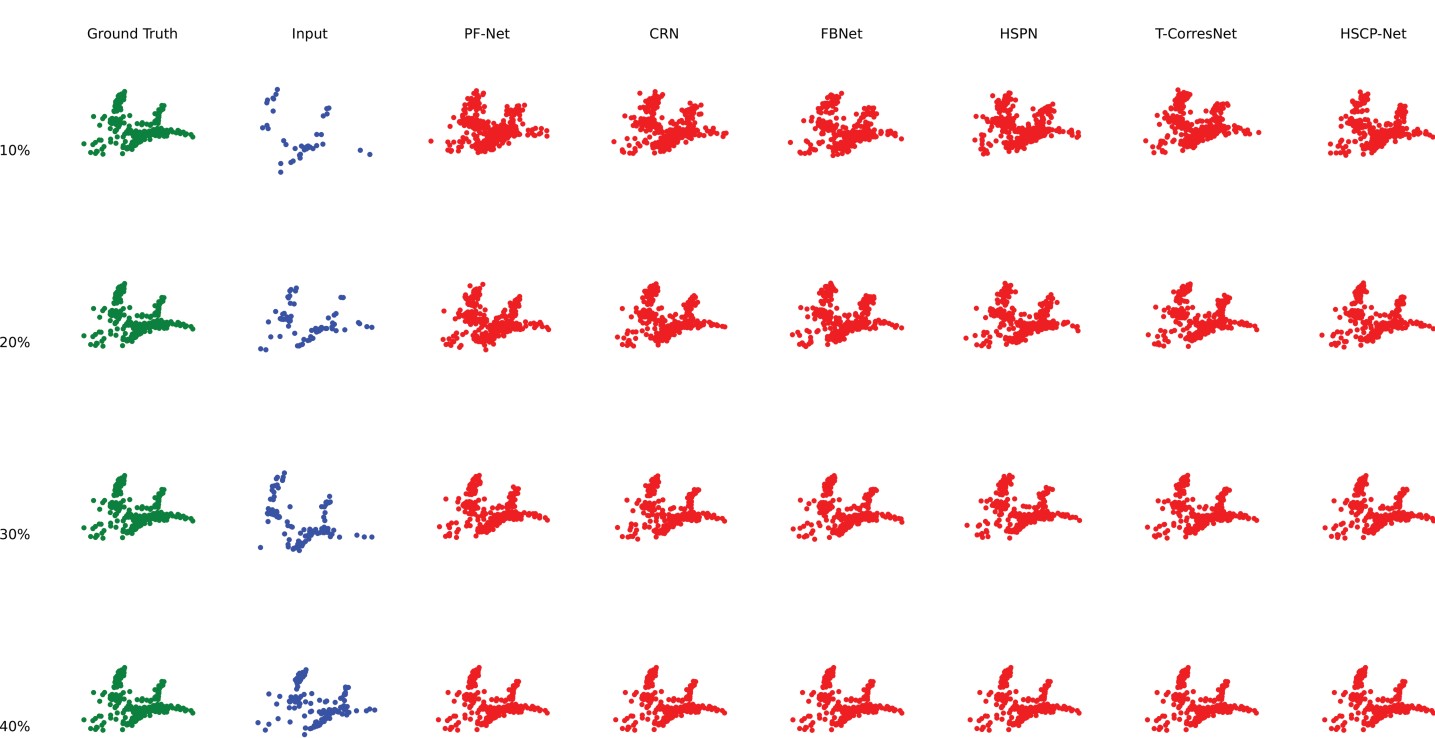

**Fig 2. Comparison of point cloud completion effects using different methods on the ShapeNet dataset.**

## Comparative experiment on mechanical parts dataset

To evaluate the performance of HSPC-Net in practical industrial scenarios, we designed a comparative experiment using a mechanical parts point cloud dataset. This dataset includes five types of mechanical parts: straight-toothed cylindrical gears, bevel gears, helical cylindrical gears, worm gears, and worm wheels. These parts exhibit high geometric complexity in their structure and have widespread applications in industrial production and inspection. The sparsity, noise, and complexity of the shapes in the point cloud data make the point cloud completion task even more challenging. The experimental results on the mechanical parts dataset are shown in Table 5.

From the experimental results, HSPC-Net demonstrates significant advantages in the point cloud completion tasks for mechanical parts, especially excelling in the SCS and PCA metrics. In the straight-toothed cylindrical gear completion task, HSPC-Net achieved the best SCS (0.915) and PCA (0.882) values, effectively recovering the fine details of the gear,

**Table 5. Category-wise comparison of point-cloud completion methods on mechanical parts.**

| Method | Spur Gear | | | Bevel Gear | | | Helical Gear | | | Worm Gear | | | Worm Wheel | | |
|---|---|---|---|---|---|---|---|---|---|---|---|---|---|---|---|
| | CD | PCA | SCS | CD | PCA | SCS | CD | PCA | SCS | CD | PCA | SCS | CD | PCA | SCS |
| PF-Net | 0.035 | 0.826 | 0.857 | 0.042 | 0.832 | 0.863 | 0.039 | 0.817 | 0.858 | 0.037 | 0.801 | 0.837 | 0.034 | 0.789 | 0.820 |
| CRN | 0.031 | 0.838 | 0.865 | 0.038 | 0.845 | 0.870 | 0.035 | 0.839 | 0.868 | 0.033 | 0.818 | 0.855 | 0.031 | 0.810 | 0.848 |
| FBNet | 0.029 | 0.854 | 0.876 | 0.036 | 0.860 | 0.881 | 0.033 | 0.855 | 0.878 | 0.031 | 0.830 | 0.861 | 0.029 | 0.820 | 0.855 |
| HSPN | 0.032 | 0.835 | 0.862 | 0.039 | 0.840 | 0.868 | 0.036 | 0.831 | 0.865 | 0.034 | 0.812 | 0.850 | 0.032 | 0.802 | 0.842 |
| T-CorresNet | 0.030 | 0.836 | 0.864 | 0.038 | 0.841 | 0.869 | 0.035 | 0.834 | 0.866 | 0.033 | 0.815 | 0.852 | 0.031 | 0.805 | 0.843 |
| HSPC-Net | **0.023** | **0.882** | **0.915** | **0.031** | **0.874** | **0.909** | **0.028** | **0.878** | **0.912** | **0.025** | **0.861** | **0.899** | **0.022** | **0.854** | **0.890** |

particularly the tooth surface morphology. Compared to CRN (SCS: 0.865, PCA: 0.838), both HSPN (SCS: 0.862, PCA: 0.835) and T-CorresNet (SCS: 0.864, PCA: 0.836) showed relatively lower performance, indicating HSPC-Net's clear superiority in recovering structural consistency and completeness. In the bevel gear completion task, HSPC-Net showed a significant improvement in PCA (0.874) compared to PF-Net (0.832), CRN (0.845), HSPN (0.840), and T-CorresNet (0.841), successfully restoring the complex geometric structures of the tooth surface and bearing parts. For the helical cylindrical gear, HSPC-Net also performed excellently in terms of geometric consistency and completion accuracy, particularly in the precise restoration of the tooth surface and slant-cut sections, with SCS (0.912) and PCA (0.878) outperforming HSPN (0.865, 0.831) and T-CorresNet (0.866, 0.834), and remaining ahead of CRN and FBNet.

Fig 3 shows the comparison of actual point cloud completion effects of different methods on the mechanical parts dataset. To be more representative, we selected the worm gear category. To keep the figure concise, we only plotted a portion of the points. The conclusions of the experiment are the same as before.

In the worm gear and worm wheel completion tasks, HSPC-Net maintained its advantages, particularly in the restoration of spiral shapes and tooth surfaces, demonstrating strong geometric consistency (worm gear SCS: 0.899, PCA: 0.861; worm wheel SCS: 0.890, PCA: 0.854). Compared to CRN (worm gear SCS: 0.855, PCA: 0.818; worm wheel SCS: 0.848, PCA: 0.810), both HSPN (0.850, 0.812; 0.842, 0.802) and T-CorresNet (0.852, 0.815; 0.843, 0.805) remained less effective. Regarding the CD metric, HSPC-Net exhibited the most precise performance on mechanical parts such as the straight-toothed cylindrical gear (0.023) and worm wheel (0.022), reflecting its geometric accuracy in point cloud completion. It outperformed

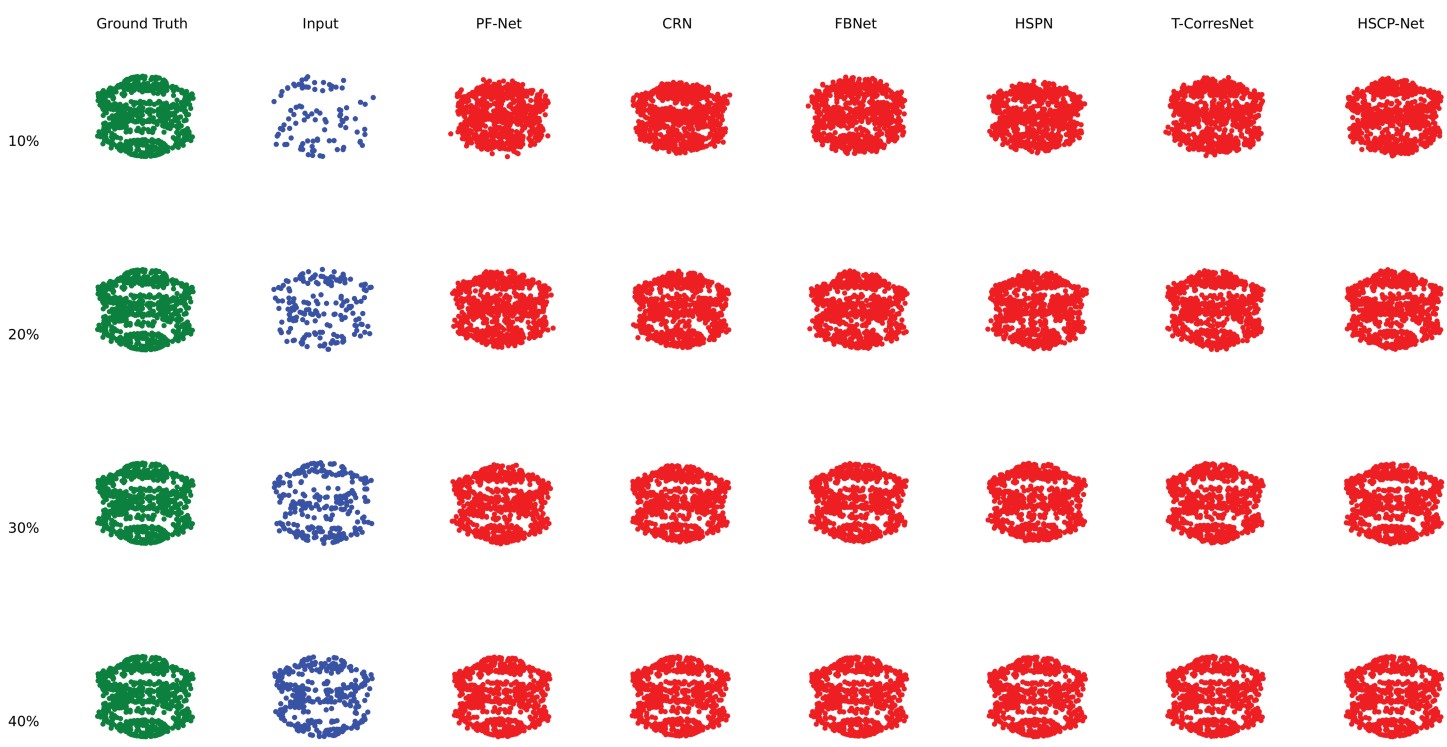

**Fig 3. Comparison of point cloud completion effects using different methods on the machine parts dataset.**

PF-Net (e.g., bevel gear CD: 0.042) and FBNet (e.g., worm gear CD: 0.031), while HSPN and T-CorresNet (e.g., bevel gear CD: 0.039 and 0.038) were also behind CRN (0.038).

These experimental results indicate that HSPC-Net, through its hierarchical structure perception and cross-modal geometric information fusion mechanisms, effectively enhances the accuracy and consistency in completing point clouds of complex-shaped mechanical parts. This suggests significant application potential, particularly in manufacturing and industrial inspection, where it can provide higher-quality point cloud completion and defect detection.

## Conclusion and future work

The proposed HSPC-Net offers a new perspective on mechanical-part point cloud completion by coupling hierarchical structure perception with cross-modal geometric fusion. Specifically, the network employs a multi-receptive field Transformer to capture structural features across scales, while integrating 2D image cues to guide the reconstruction of missing geometry. Experiments confirm that this design delivers higher completion accuracy and robustness, especially for parts with intricate geometries. Nevertheless, two limitations should be acknowledged. First, our quantitative study compares HSPC-Net only with five representative state-of-the-art methods; the domain is evolving rapidly, and a broader benchmark will be necessary to capture the latest advances. Second, we have not yet evaluated the model on the full ShapeNet dataset or on third-party mechanical-component benchmarks, which restricts conclusions about cross-dataset generalization. Future work will therefore extend the comparison to a larger pool of recent baselines and conduct comprehensive testing on both the complete ShapeNet corpus and additional industrial point-cloud collections. In parallel, we plan to investigate more efficient fusion schemes as well as multi-task or self-supervised objectives to further boost HSPC-Net's adaptability to large-scale, multi-modal industrial scenarios.

## Acknowledgments

The authors would like to acknowledge all contributors to this work, all of whom are included in the author list.

## Author contributions

**Supervision:** Honghui Fan, Hongjin Zhu.

**Validation:** Hongjin Zhu.

**Writing – original draft:** Yuchao Jiang, Honghui Fan.

**Writing – review & editing:** Yuchao Jiang, Honghui Fan.

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
