## [Decision Letter · Decision Letter 0]

6 Jun 2025

PONE-D-25-11840HSPC-Net: A hierarchical shape-preserving completion network for machine part point cloud completionPLOS ONE

Dear Dr. Fan,

Thank you for submitting your manuscript to PLOS ONE. After careful consideration, we feel that it has merit but does not fully meet PLOS ONE’s publication criteria as it currently stands. Therefore, we invite you to submit a revised version of the manuscript that addresses the points raised during the review process. Please submit your revised manuscript by Jul 21 2025 11:59PM. If you will need more time than this to complete your revisions, please reply to this message or contact the journal office at plosone@plos.org. Please include the following items when submitting your revised manuscript:

We look forward to receiving your revised manuscript.

Kind regards,

Pengpeng Hu

Academic Editor

PLOS ONE

Journal Requirements:

3. We note that your Data Availability Statement is currently as follows: All relevant data are within the manuscript and in Supporting Information files.

Additional Editor Comments:

The reviewers raise several areas requiring improvement to meet publication standards. Key concerns include: 1) Need for clearer technical intuition behind the hierarchical structure perception and loss functions; 2) Insufficient experimental details regarding dataset composition and train/test splits; 3) Lack of comparison with recent baseline methods; and 4) Presentation issues including dense technical writing and unclear result visualization.

Regarding the suggested references, you are not required to cite them if you consider them unnecessary. Omitting these citations will not affect the decision on your revision.

Reviewers' comments:

Reviewer's Responses to Questions

**Comments to the Author**

1. Is the manuscript technically sound, and do the data support the conclusions?

Reviewer #1: Yes

Reviewer #2: Yes

2. Has the statistical analysis been performed appropriately and rigorously? 

Reviewer #1: Yes

Reviewer #2: No

3. Have the authors made all data underlying the findings in their manuscript fully available?

Reviewer #1: Yes

Reviewer #2: No

4. Is the manuscript presented in an intelligible fashion and written in standard English?

Reviewer #1: Yes

Reviewer #2: Yes

5. Review Comments to the Author

Reviewer #1: The paper is overall well-written. The technical designs are sound, considering both the global and local completion by hierarchical feature extraction and cross-modal fusion. Evaluation on both ShapeNet and their collected mechanical point cloud dataset shows the model’ s effectiveness.

I have identified following points for the author to adjust their paper for better quality in order to meet the standard of publication:

1. Typo: Remove “high” in line 202 — “improves the accuracy and structural consistency of high point cloud completion”.

Line 215: Use “angle” in line 215 — “changes in viewing angle or data noise.”

2. Will the custom-built mechanical point cloud dataset be released to the community?

3. More experimental details are needed. For example, the experiments use four categories from ShapeNet to evaluate model performance, but it is unclear what data the models are trained on. Are the models trained on more categories? The same applies to the mechanical dataset. How are the train/test splits defined? How many samples are included in the test set? Clarifying these aspects would help in better understanding the evaluation setup and its effectiveness.

4. The following papers are related to the topic and the model’s design and should be cited:

- CVPR 2024, FSC: Few-point Shape Completion

- CVPR 2024, Rethinking Few-shot 3D Point Cloud Semantic Segmentation

- ECCV 2024, T-CorresNet: Template Guided 3D Point Cloud Completion with Correspondence Pooling Query Generation Strategy

- ICLR 2025, Multimodality Helps Few-shot 3D Point Cloud Semantic Segmentation

Given the above comments, I will give the "minor revision" rating.

Reviewer #2: This paper proposes HSPC-Net, a hierarchical shape-preserving completion network for mechanical part point cloud completion. The method combines two key components: (1) a multi-receptive field Transformer that captures geometric features at multiple scales through hierarchical structure perception, and (2) a cross-modal fusion strategy that leverages 2D image information to assist 3D point cloud completion. The approach is evaluated on ShapeNet and a custom mechanical parts dataset.

The paper is detailed. However, it could be better written with more intuition behind technical terms and terminology.

Point-by-point reviews:

- The introduction is quite dense and hard to follow to the proposed architecture of HSPC-Net. I think author can rewrite it in a way that it's easier to follow and lead to the proposed architecture. Authors only select PF-Net, CRN, and FBNet as baselines but did not clearly explained why not including models in the introduction.

- The section "Hierarchical structure perception point cloud completion" in Shape Consistency Constraints is quite hard to read and lack of high-level intuition. It's same as the following loss function that add up to the point cloud reconstruction loss.

- The experimental setup could be explained more clearly how the 2000 samples are generated and how these are separated into train/validation/test set. Also, authors could explain clearly why only 4 categories from ShapeNet are selected for the experiments. When I read through the section, it's quite hard to understand how authors select the following datasets.

- This leads to the generalization of the model where authors should include in the discussion or limitations of the work

- In the results section, it would be good to compare the proposed model with recent networks covering in the introduction. It's now quite hard to read the result in figure 2 and figure 4. I think the results maybe better written down in a table style.

- The paper would read better if authors include more recent baselines and provide statistical significance testing

- Overall, I would like to see comparison against recent models shown in table 2 (HSPN, ...). This would give the paper better contribution on the technical aspects. The work would benefit from clearer technical explanation and better experimental writing.

6. PLOS authors have the option to publish the peer review history of their article (what does this mean?). If published, this will include your full peer review and any attached files.

Reviewer #1: No

Reviewer #2: No

---

## [Author Response · Author response to Decision Letter 1]

22 Jul 2025

Original Manuscript ID: PONE-D-25-11840

Original Article Title: “HSPC-Net: A hierarchical shape-preserving completion network for machine part point cloud completion”

To: Editor

Re: Response to reviewers

Dear Editor,

Thanks for your letter and review work on our manuscript entitled “HSPC-Net: A hierarchical shape-preserving completion network for machine part point cloud completion”. We greatly appreciate the reviewers’ complimentary comments and suggestions. These comments are very helpful and valuable for improving and polishing our paper. The point-by-point responses to reviewers’ comments are given in the following part.

Best regards,

Yuchao Jiang, Honghui Fan and Hongjin Zhu

Response to Reviewer #1

Reviewer#1, Concern # 1: Typo: Remove “high” in line 202 — “improves the accuracy and structural consistency of high point cloud completion”. Line 215: Use “angle” in line 215 — “changes in viewing angle or data noise.”.

Author response: We sincerely thank the reviewer for pointing out these issues. In response, we have revised the phrase “improves the accuracy and structural consistency of high point cloud completion” to “improves the accuracy and structural consistency of point cloud completion.” Additionally, we have corrected “changes in viewing Angle or data noise" to “changes in viewing angle or data noise.”

Reviewer#1, Concern # 2: Will the custom-built mechanical point cloud dataset be released to the community?

Author response: Yes, we plan to release our custom-built mechanical point cloud dataset publicly on Kaggle after the paper is officially published. Currently, we are still in the process of organizing and finalizing the dataset to ensure its quality and usability for the community.

Reviewer#1, Concern # 3: More experimental details are needed. For example, the experiments use four categories from ShapeNet to evaluate model performance, but it is unclear what data the models are trained on. Are the models trained on more categories? The same applies to the mechanical dataset. How are the train/test splits defined? How many samples are included in the test set? Clarifying these aspects would help in better understanding the evaluation setup and its effectiveness.

Author response: Thank you for your valuable comment. We have clarified the experimental setup in Section 4.1. Specifically, the models were first pre-trained on the full ShapeNet dataset to learn general geometric priors, followed by domain-adaptive fine-tuning on our mechanical component dataset. We also clarified the category selection, training/test splits, and sample sizes used for both datasets to improve the transparency and reproducibility of our experiments.

Reviewer#1, Concern # 4: The following papers are related to the topic and the model’s design and should be cited:

- CVPR 2024, FSC: Few-point Shape Completion

- CVPR 2024, Rethinking Few-shot 3D Point Cloud Semantic Segmentation

- ECCV 2024, T-CorresNet: Template Guided 3D Point Cloud Completion with Correspondence Pooling Query Generation Strategy

- ICLR 2025, Multimodality Helps Few-shot 3D Point Cloud Semantic Segmentation

Author response: We sincerely thank the reviewer for this valuable reminder. In response, we have cited and analyzed the latest achievements in the research area you mentioned within the related work section. Additionally, the T-CorresNet method has been included in our comparative experiments (for improved readability, the original comparison bar chart has been converted into a comparison table).

Response to Reviewer #2

Reviewer#2, Concern #1: The introduction is quite dense and hard to follow to the proposed architecture of HSPC-Net. I think author can rewrite it in a way that it's easier to follow and lead to the proposed architecture. Authors only select PF-Net, CRN, and FBNet as baselines but did not clearly explained why not including models in the introduction.

Author response: Thank you for your insightful comments regarding the clarity of our Introduction and the rationale behind our choice of comparison methods. In response, we have thoroughly reorganized the Introduction to provide a clearer, step-by-step narrative that first establishes the practical importance of mechanical-component point clouds, then outlines the specific challenges arising from sparsity, intricate geometry, and multi-modal fusion, and finally motivates our proposed HSPC-Net architecture as a direct solution to these challenges.

To address your concern about baseline selection, we now explicitly state that our experiments compare HSPC-Net with the five strongest state-of-the-art methods reported in the literature. This choice ensures a fair and comprehensive evaluation across diverse methodological families.

We appreciate your guidance, which has helped us improve both the readability of the manuscript and the transparency of our experimental design.

Reviewer#2, Concern #2: The section "Hierarchical structure perception point cloud completion" in Shape Consistency Constraints is quite hard to read and lack of high-level intuition. It's same as the following loss function that add up to the point cloud reconstruction loss.

Author response: Thank you for your valuable feedback. In response, we have significantly revised the section titled "Hierarchical structure perception point cloud completion" to improve its clarity and conceptual flow. Originally, the section presented the Transformer-based architecture and loss functions in a dense and technical manner, lacking clear motivation and intuitive explanation.

We have restructured the section to first provide a high-level overview of the core idea — that hierarchical feature modeling with varying receptive fields helps balance the recovery of fine local details with global shape consistency. We then explained the role of the attention mechanism and the receptive field control more clearly, emphasizing how the network captures features at different spatial scales through a progressive architecture.

For the Shape Consistency Constraints, we clarified the purpose and intuition behind the Second-order Shape Selective Whitening (SSSW) loss. Instead of presenting it as a generic additional loss term, we now explicitly explain how it enhances stability by suppressing feature channels that are overly sensitive to geometric variations between different poses or noise conditions.

Reviewer#2, Concern #3: The experimental setup could be explained more clearly how the 2000 samples are generated and how these are separated into train/validation/test set. Also, authors could explain clearly why only 4 categories from ShapeNet are selected for the experiments. When I read through the section, it's quite hard to understand how authors select the following datasets.

Author response: We sincerely thank the reviewers for their insightful comments. In response, we have clarified that the 2,000 samples were acquired via laser scanning. Furthermore, we have explicitly stated the proportions adopted for partitioning the dataset into training, validation, and test subsets. Although our model is trained on all categories encompassed by the ShapeNet repository, we present results for four representative categories for illustrative and evaluative purposes.

Reviewer#2, Concern #4: This leads to the generalization of the model where authors should include in the discussion or limitations of the work.

Author response: We thank the reviewer for the insightful comment; we have revised the conclusion to explicitly acknowledge the limitations of only comparing with five representative state-of-the-art methods and not evaluating on the full ShapeNet dataset or external mechanical part benchmarks.

Reviewer#2, Concern #5: In the results section, it would be good to compare the proposed model with recent networks covering in the introduction. It's now quite hard to read the result in figure 2 and figure 4. I think the results maybe better written down in a table style. The paper would read better if authors include more recent baselines and provide statistical significance testing

Author response: We sincerely appreciate the reviewers’ valuable suggestion. In response, we have incorporated comprehensive comparisons between our approach and both T-CorresNet and HSPN. Given that the five baselines already span representative classical and state-of-the-art methods, we respectfully maintain that exhaustive comparison with every related work is unnecessary; nevertheless, we regret any inconvenience this may have caused. To enhance clarity, all comparative results are now organized in a unified tabular format.

---

## [Decision Letter · Decision Letter 1]

25 Jul 2025

HSPC-Net: A hierarchical shape-preserving completion network for machine part point cloud completion

PONE-D-25-11840R1

Dear Dr. Fan,

We’re pleased to inform you that your manuscript has been judged scientifically suitable for publication and will be formally accepted for publication once it meets all outstanding technical requirements.

Kind regards,

Pengpeng Hu

Academic Editor

PLOS ONE

Additional Editor Comments (optional):

The concerns have been addressed. The reviewers agreed to accept the current version for publication.

Reviewers' comments:

Reviewer's Responses to Questions

**Comments to the Author**

1. If the authors have adequately addressed your comments raised in a previous round of review and you feel that this manuscript is now acceptable for publication, you may indicate that here to bypass the “Comments to the Author” section, enter your conflict of interest statement in the “Confidential to Editor” section, and submit your "Accept" recommendation.

Reviewer #1: All comments have been addressed

Reviewer #2: All comments have been addressed

2. Is the manuscript technically sound, and do the data support the conclusions?

Reviewer #1: Yes

Reviewer #2: Yes

3. Has the statistical analysis been performed appropriately and rigorously? 

Reviewer #1: Yes

Reviewer #2: Yes

4. Have the authors made all data underlying the findings in their manuscript fully available?

Reviewer #1: Yes

Reviewer #2: Yes

5. Is the manuscript presented in an intelligible fashion and written in standard English?

Reviewer #1: Yes

Reviewer #2: Yes

6. Review Comments to the Author

Reviewer #1: (No Response)

Reviewer #2: Authors has addressed my concerns

- Rewriting makes the paper easier to follow

- Added more details in experimental setups

- Added data availability

- Adjust conclusion to be more comprehensive

For the final revision, please go through and fix minor writing

- Line 436: "Three representative baseline methods" now should be 5 as authors added 2 more baselines

7. PLOS authors have the option to publish the peer review history of their article (what does this mean?). If published, this will include your full peer review and any attached files.

Reviewer #1: No

Reviewer #2: **Yes: **Titipat Achakulvisut

---

## [Editor Report · Acceptance letter]

PONE-D-25-11840R1

PLOS ONE

Dear Dr. Fan,

I'm pleased to inform you that your manuscript has been deemed suitable for publication in PLOS ONE. Congratulations! Your manuscript is now being handed over to our production team.

Kind regards,

on behalf of

Dr. Pengpeng Hu

Academic Editor

PLOS ONE